# Genomic Alterations Associated with Estrogen Receptor Pathway Activity in Metastatic Breast Cancer Have a Differential Impact on Downstream ER Signaling

**DOI:** 10.3390/cancers15174416

**Published:** 2023-09-04

**Authors:** Lindsay Angus, Marcel Smid, Saskia M. Wilting, Manouk K. Bos, Neeltje Steeghs, Inge R. H. M. Konings, Vivianne C. G. Tjan-Heijnen, Johanna M. G. H. van Riel, Agnes J. van de Wouw, Edwin Cuppen, Martijn P. Lolkema, Agnes Jager, Stefan Sleijfer, John W. M. Martens

**Affiliations:** 1Department of Medical Oncology, Erasmus MC Cancer Institute, Erasmus University Medical Cancer, Dr. Molewaterplein 40, 3015 GD Rotterdam, The Netherlands; m.smid@erasmusmc.nl (M.S.); s.wilting@erasmusmc.nl (S.M.W.); m.k.bos@erasmusmc.nl (M.K.B.); m.lolkema@erasmusmc.nl (M.P.L.); a.jager@erasmusmc.nl (A.J.); s.sleijfer@erasmusmc.nl (S.S.); j.martens@erasmusmc.nl (J.W.M.M.); 2Department of Medical Oncology, The Netherlands Cancer Institute, 1066 CX Amsterdam, The Netherlands; n.steeghs@nki.nl; 3Center for Personalized Cancer Treatment, 6500 HB Nijmegen, The Netherlands; vcg.tjan.heijnen@mumc.nl (V.C.G.T.-H.);; 4Department of Medical Oncology, Cancer Center Amsterdam, Amsterdam UMC, Vrije Universiteit Amsterdam, 1081 HV Amsterdam, The Netherlands; i.konings@amsterdamumc.nl; 5Department of Medical Oncology, GROW-School for Oncology and Developmental Biology, Maastricht University Medical Center, 6229 HX Maastricht, The Netherlands; 6Department of Internal Medicine, Elisabeth-TweeSteden Hospital, 5022 GC Tilburg, The Netherlands; jmgh.vanriel@etz.nl; 7Department of Medical Oncology, VieCuri Medical Center, 5912 BL Venlo, The Netherlands; yvdwouw@viecuri.nl; 8Center for Molecular Medicine and Oncode Institute, University Medical Center Utrecht, 3584 CX Utrecht, The Netherlands; e.cuppen@hartwigmedicalfoundation.nl; 9Hartwig Medical Foundation, 1098 XH Amsterdam, The Netherlands

**Keywords:** breast cancer, whole genome sequencing, RNA sequencing, endocrine resistance

## Abstract

**Simple Summary:**

Breast cancer patients often receive anti-hormonal treatment if their tumor is positive for the Estrogen Receptor (ER), but tumors may become resistant to this therapy and still metastasize. We studied 101 of such metastatic lesions and investigated these lesions for mutated genes and mutation patterns, in combination with the level of expression of relevant genes. Our aim was to better understand the mechanisms that are involved in the resistance to anti-hormonal treatment. The analyses showed two distinct groups of patients, each with specific mutations. One group clearly showed an ongoing, active ER and its associated signal route; these patients probably still would benefit from ER-targeting agents. We advocate for combining mutation and expression analyses on metastatic lesions, to maximize the group of patients that still may benefit from existing or new anti-hormonal treatments targeting ER or its signaling network.

**Abstract:**

Mutations in the estrogen receptor gene (*ESR1*), its transcriptional regulators, and the mitogen-activated protein kinase (MAPK) pathway are enriched in patients with endocrine-resistant metastatic breast cancer (MBC). Here, we integrated whole genome sequencing with RNA sequencing data from the same samples of 101 ER-positive/HER2-negative MBC patients who underwent a tumor biopsy prior to the start of a new line of treatment for MBC (CPCT-02 study, NCT01855477) to analyze the downstream effects of DNA alterations previously linked to endocrine resistance, thereby gaining a better understanding of the associated mechanisms. Hierarchical clustering was performed using expression of *ESR1* target genes. Genomic alterations at the DNA level, gene expression levels, and last administered therapy were compared between the identified clusters. Hierarchical clustering revealed two distinct clusters, one of which was characterized by increased expression of *ESR1* and its target genes. Samples in this cluster were significantly enriched for mutations in *ESR1* and amplifications in *FGFR1* and *TSPYL.* Patients in the other cluster showed relatively lower expression levels of *ESR1* and its target genes, comparable to ER-negative samples, and more often received endocrine therapy as their last treatment before biopsy. Genes in the MAPK-pathway, including *NF1*, and *ESR1* transcriptional regulators were evenly distributed. In conclusion, RNA sequencing identified a subgroup of patients with clear expression of *ESR1* and its downstream targets, probably still benefiting from ER-targeting agents. The lower ER expression in the other subgroup might be partially explained by ER activity still being blocked by recently administered endocrine treatment, indicating that biopsy timing relative to endocrine treatment needs to be considered when interpreting transcriptomic data.

## 1. Introduction

Breast cancer is the most common cancer among women worldwide [1]. Although the majority of breast cancer patients are cured, 20–30% of patients will develop incurable metastatic disease [2]. As 60–70% of patients with metastatic breast cancer (MBC) have tumors expressing estrogen receptor alpha (ER), endocrine therapy has become the mainstay treatment for these patients [3]. Despite the success of endocrine treatment, unfortunately, 20–30% [4,5,6,7] of patients have no clinical benefit from first-line endocrine therapy due to intrinsic resistance, whereas the remainder of initially responding patients will eventually develop resistance during therapy. Once resistant, tumors become more aggressive and more difficult to treat.

Recently performed sequencing efforts on metastatic tumor biopsies have revealed several mechanisms conferring resistance against endocrine treatment [8,9,10]. Activating mutations in the ligand-binding domain of the gene encoding the ER, *ESR1,* lead to constitutive activity of ER and have been related to shorter progression-free survival on single-agent aromatase inhibitors (AIs) [11,12]. After exposure to nonsteroidal AIs for metastatic disease, 29–39% [11,12,13] of patients harbor *ESR1* mutations, whereas *ESR1* mutations are quite rare in primary tumors (only ~3% of patients [14,15]) and newly diagnosed metastatic disease after adjuvant treatment with AIs (5.3–6.4% of patients [16,17,18]). The most common *ESR1* mutations have been functionally annotated whereby D538G and Y537 are known for their constitutive activity, whereas the E380Q variant “only” leads to estrogen hypersensitivity [15].

Additionally, mutations in the mitogen-activated protein kinase (MAPK) pathway are enriched in MBC as well, including alterations in *ERBB2*, *NF1*, *EGFR*, *ERBB3*, *KRAS, BRAF*, *MAP2K1*, and *HRAS* [8]. Of these genes, inactivating mutations in *NF1* are most frequently observed and are mutually exclusive with activating mutations in *ESR1* [8,19]. Recently, Zheng et al. have shown that *NF1* acts as a co-repressor of ER-α transcription, so when *NF1* gets inactivated, this leads to an increased expression of ER target genes such as *GREB1* and *TFF1* [20]. In addition to mutations in *ESR1* and MAPK pathway genes, alterations in ER transcriptional regulators have been associated with endocrine resistance as well, including *MYC*, *FOXA1*, *CTCF*, and *TBX3* [8].

Although whole genome and whole exome sequencing (WGS and WES, respectively) efforts have revealed enriched gene alterations in metastatic tumors compared to primary breast cancer, integration with gene expression is necessary to enable subsequent analysis of the downstream effects of these alterations. Here, we integrated WGS data with matched RNA sequencing data obtained from biopsies of 101 patients with ER-positive/HER2-negative metastatic breast cancer to assess the relation between *ESR1* mutations, alterations in *ESR1*-transcriptional regulators, and MAPK pathway mutations and the activity of the ER pathway. Samples and associated genomic features were ordered by hierarchical clustering using RNA expression of ER pathway genes to get a better understanding on how these mutations associate with *ESR1* expression and its target genes. Moreover, clinical data such as type of treatment prior to tissue biopsy—especially focusing on the major types of endocrine therapy such as selective estrogen receptor modulators such as tamoxifen, aromatase inhibitors, and estrogen receptor degraders such as fulvestrant—were associated with the expression of ER target genes. Finally, we performed an exploratory analysis to correlate the tumors’ transcriptome with subsequent best response to endocrine therapies.

## 2. Methods

### 2.1. Study Design and Patients

For the current analyses, we selected patients with metastatic breast cancer who were included under the protocol of the Center for Personalized Cancer Treatment (CPCT) consortium (CPCT-02 Biopsy Protocol, ClinicalTrial.gov no. NCT01855477), which was approved by the medical ethics committee of the University Medical Center Utrecht, the Netherlands. The consortium and the whole patient cohort have been described in detail recently [10,21]. Briefly, patients of ≥18 years old, with incurable locally advanced or metastatic solid tumors, from whom a histological biopsy could be safely obtained and systemic treatment with anticancer agents was indicated were eligible for inclusion. Biopsies of metastatic lesions from patients with ER-positive/HER2-negative breast cancer (obtained via pathology reports of the primary tumor), from which both WGS and RNA sequencing data were available, were included (n = 101). A cohort of 63 patients with ER-negative metastatic breast cancer was included as readout for low/absent ER expression.

### 2.2. Prior Endocrine Therapy

As we investigated the expression of estrogen-regulated genes, it is important to establish in which patients expression of these genes was potentially influenced by endocrine therapy. We expected that endocrine therapies that were given as the last systemic therapy before the tissue biopsy was taken could still impact the expression of estrogen regulated genes. Therefore, we registered the last treatment that was administered before biopsy and grouped these treatments as follows: (1) tamoxifen; (2) aromatase inhibitors; (3) fulvestrant; (4) chemotherapy or other non-endocrine therapy; (5) combination endocrine therapy (endocrine backbone combined with either CDK-4/6 inhibitors or everolimus); (6) no therapy within one year before the biopsy.

### 2.3. Treatment Outcome and Response to Endocrine Therapies

Tumor responses were evaluated according to RECIST v1.1 every 8–12 weeks of treatment, and the best overall response was defined as complete response (CR), partial response (PR), stable disease (SD), or progressive disease (PD) [22]. For the subset of patients who started with endocrine therapy after their tumor biopsy, RNA profiles were associated with the best overall response. Patients without response information or patients who did not start with endocrine therapy after the investigated biopsy were excluded from this analysis.

### 2.4. WGS and Data Analyses

Genomic features, such as somatic single nucleotide variants and copy number alterations, were extracted from WGS data as previously described [10]. Mutational signatures v3 [23] were called using R package MutationalPatterns v1.10.0 [24], focusing on single and double base signatures. Alterations in genes within the same pathway were grouped based on the findings of Razavi et al. [8]. In short, we used the following definitions: ESR1 hotspot mutations: mutations in codons 536–538 and 380; MAPK pathway alterations: mutations in *BRAF*, *ERBB3*, *HRAS*, *KRAS, MAP2K1,* mutations and deep gains in *EGFR* and *ERBB2*, and mutations and deep deletions in *NF1*; ER transcriptional regulators: mutations in *CTCF* and *TBX3*, mutations and deep gains in *FOXA1* and deep gains of *MYC*.

### 2.5. RNA Sequencing: RNA Isolation, Library Preparation, and Sequencing

RNA was isolated from fresh frozen tissue biopsies using the QiaSympony RNA kit (Qiagen, Venlo, the Netherlands) as per manufacturer’s instructions and quantified using the Qubit RNA IQ Assay (Invitrogen, Life Technologies, Carlsbad, CA, USA) according to the manufacturer’s instructions using the Qubit fluorometer (Invitrogen). The RNA yield from the tissue biopsies ranged between 50 and 5000 ng. Library preparation was performed using the KAPA RNA HyperPrep kit (Roche, Tokyo, Japan) with RiboErase (Human/Mouse/Rat) on an automated liquid handing platform (Beckman Coulter, Pasadena, CA, USA) using a total of 50–100 ng RNA. RNA was fragmented at 85 °C for 6 min in the presence of magnesium to a target fragment length of 300 bp. Barcoded libraries were sequenced as pools on NextSeq 500 (V2.5 reagents) generating 2 × 75 read pairs or, at a later stage, on a NovaSeq 6000 generating 2 × 150 read pairs using standard settings (Illumina, Tokyo, Japan). Binary base call (BCL) output from the sequencing platform was converted to FASTQ using Illumina bcl2fastq tool (versions 2.17 to 2.20 have been used) using default parameters

### 2.6. Processing of RNA Sequencing Data

Next, sequence reads in the FASTQ files were trimmed for adapter sequences using fastp v0.20.0 [25]. The resulting FASTQ files were mapped to GRCh38 using STAR v2.6.1d9 [26], and Sambamba v0.7.0 [27] was used to mark duplicates and index the resulting BAM files. Gene annotation was derived from GENCODE Release 30 (https://www.gencodegenes.org/). To obtain gene expression levels (raw read counts), featureCounts v1.6.3 [28] was used. Finally, the count matrix was normalized using the GeTMM method [29], using R v3.6.0 [30].

### 2.7. Hierarchical Clustering on RNA Expression Levels of ER-Regulated Genes

Gene expression values were available as log2 values for 19,986 protein coding transcripts. Since unsupervised clustering of the top 5000 variable genes was driven largely by the site of metastatic biopsy—i.e., all liver biopsies clustered together (Appendix A), we corrected for this by performing a ComBat correction [31]. After correction, we performed hierarchical clustering of the samples using our defined set of *ESR1* target genes: *AP1B1*; *CA12*; *CDH26*; *CELSR2*; *COL18A1*; *COX7A2L*; *CTSD*; *DSCAM*; *EBAG9*; *ERBB2*; *ESR1*; *GREB1*; *HSPB1*; *IGFBP4*; *KRT19*; *MYC*; *NRIP1*; *PGR*; *PISD*; *PTMA*; *RARA*; *SGK3*; *SOD1*; *TFF1*; *TRIM25*; *CCN5*; *XBP1* [32]. Expression levels were first median-centered before clustering and then hierarchically clustered using average linkage and uncentered correlation distance metric [33] and visualized using Treeview [34].

### 2.8. Statistical Methods

Pearson’s chi-squared test or Fisher’s exact test (in case of too few expected events) was used to evaluate categorical data. To compare continuous variables, a Mann–Whitney *U* test or a Kruskal-Wallis H test was performed. All statistical tests were considered statistically significant at a two-sided *p* < 0.05. Stata 13.0 (StataCorp, College Station, TX, USA) and R v.3.6.0. were used for statistical analyses. We used the Benjamini–Hochberg procedure to correct *p* values for multiple hypothesis testing when appropriate.

## 3. Results

### 3.1. Patient Cohort

We integrated prospectively collected WGS and RNA sequencing data from tissue biopsies of 101 patients with ER–positive/HER2–negative metastatic breast cancer (Table 1). Seventy-six (75%) patients received one or more prior treatments. Focusing on the treatment last administered before biopsy, 8 patients received tamoxifen, 20 an AI, 2 fulvestrant, 6 combination therapy with endocrine backbone, 22 chemotherapy or other non-endocrine-containing therapy, and 43 received no prior treatment within one year before the biopsy.

Regarding the alterations in pathways associated with endocrine resistance, 16 patients (16%) had a mutation in *ESR1*, of which 15 were located in the ligand-binding domain (p.D538G (n = 8); p.Y537S (n = 4); p.Y537N (n = 1); p.L536P (n = 1); p.E380Q (n = 1)), and one was a nonsense mutation located in the activation function 1 domain (p.Q17 *). Next to *ESR1* mutations, 27 patients (27%) had alterations at the DNA level in the MAPK pathway and 45 (45%) patients in the ER transcriptional regulators (Figure 1).

### 3.2. Hierarchical Clustering Reveals Distinct RNA Expression of ER Target Genes between ESR1 Mutant and ESR1 Wild-Type Samples

The transcriptomic data were first corrected to adjust for a bias in the biopsy site (Appendix A). After ComBat correction [31], we verified via unsupervised clustering that the biopsy site was distributed over the clusters. Next, we used our defined set of ER target genes (see methods for gene list) and verified that *ESR1* expression and the average expression of the ER target genes were not associated with biopsy site (Kruskal-Wallis *p* = 0.390 and *p* = 0.734, respectively). Subsequently, we performed hierarchical clustering (Figure 2) using the gene expression levels of ER target genes (see methods for gene list) of 101 metastatic lesions, which revealed two clusters of samples. Compared to cluster B, samples in cluster A (n = 47) had a higher average gene expression of all ER target genes (Mann–Whitney *p* = 4.95 × 10^−17^) (Figure 3, right panel). *ESR1* itself (Mann–Whitney *p* = 8.11 × 10^−7^, Figure 3, left panel) and known ER target genes such as *GREB1* (Mann–Whitney *p* = 1.32 × 10^−16^) and *PGR* (Mann–Whitney *p* = 6.44 × 10^−12^) also had a higher expression in cluster A than in cluster B. We validated this by evaluating the *ESR1* module [35], a gene signature associated with an active ER pathway. Appendix A shows a significant higher (*p* = 1.8 × 10^−5^, Mann–Whitney) module score in samples from cluster A.

As all samples were derived from patients with ER-positive primary breast cancer, we investigated whether the average gene expression of the ER target genes in cluster B were at the same level as ER-negative samples. Comparing samples from cluster B (n = 54) with 63 ER-negative metastatic breast cancer samples, we found that the *ESR1* expression itself was higher in samples from cluster B (Figure 3, left panel). However, the average expression level of the ER target genes was similar in the samples in cluster B and in ER-negative samples (Figure 3, right panel).

As cluster A represented samples with higher expression of *ESR1* and its target genes, we investigated whether specific underlying genomic alterations were enriched in cluster A versus cluster B. First, we focused on genes and gene pathways previously associated with endocrine resistance: *ESR1* mutations, *ESR1*-transcriptional regulators, and MAPK pathway alterations. Cluster A was characterized by an enrichment of samples with hotspot mutations in *ESR1* (Fisher’s exact, *p* < 0.001) (p.D538G n = 8; p.Y537S n = 4; p.Y537N (n = 1); and p.E380Q (n = 1)). Cluster B contained only two samples with an *ESR1* mutation; one harbored a truncating mutation p.Q17*, and the other one contained mutation p.L536P. However, the first mutation is not located in the ligand-binding domain and, importantly, is an inactivating mutation. For the latter variant, constitutive activity has been shown, but in this one patient the “ER-low” profile was found [15]. Overall, these results support the observation that activating *ESR1* mutations are linked to a distinct profile with increased expression of *ESR1* and ER target genes, as was the case in cluster A. Alterations in *ESR1* transcriptional regulator genes and MAPK pathway genes were equally distributed over the two clusters. Considering *NF1* and the RAS pathway genes (*BRAF/HRAS/KRAS*) separately, these seemed more frequently affected in cluster B, but this difference was not statistically significant (Appendix A). Since tumors with *PIK3CA* mutations recently have been shown to yield sensitivity to alpelisib, a PI3K inhibitor [36], we compared its frequency between both clusters and observed that cluster B (n = 54) was characterized by a modest but significant enrichment of *PIK3CA* mutations (Pearson chi-square, *p* = 0.029).

Furthermore, we compared the relative contributions of mutational signatures between clusters A and B and observed that the contribution of COSMIC mutational signature 3 (associated with homologous recombination deficiency) was higher in cluster A than in B (Mann–Whitney *p* = 0.011). Conversely, COSMIC mutational signature 2, associated with APOBEC mutagenesis, was enriched in cluster B (Mann–Whitney *p* = 0.011). Since we expected that endocrine therapies that were discontinued just before the tissue biopsy was taken could still impact the expression of estrogen-regulated genes, we compared the last administered treatment versus the clusters. The type of last administered treatment was different between both clusters (Fisher’s exact, *p* = 0.016), and we observed that patients in cluster B had more frequently received an AI or tamoxifen directly prior to their biopsy, whereas samples from patients in cluster A more frequently received chemotherapy or no treatment at all within one year prior to biopsy. For patients who received an AI prior to their tissue biopsy (n = 20), 8 were still on treatment while the biopsy was taken, and the other 12 patients stopped treatment 3–186 days prior to the biopsy. Of all 20 patients who received an AI prior to their biopsy, 4 patients clustered in cluster A. These 4 patients stopped with the AI 3–18 days prior to biopsy. For patients who received tamoxifen prior to their tissue biopsy (n = 8), 2 patients were still on treatment when the biopsy was taken, and the other 6 patients stopped 3–93 days prior to their biopsy. Of all 8 patients who received tamoxifen prior to their biopsy, 1 patient clustered in cluster A; this patient stopped with tamoxifen on the day of the biopsy.

### 3.3. FGFR1 and TSPYL Amplifications Are Enriched beyond ESR1 Mutations

To investigate whether other DNA alterations, next to *ESR1* mutations, were enriched in cluster A that could be linked to the high expression of *ESR1* and its target genes, we compared samples that were *ESR1* wild-type from cluster A with all samples from cluster B. First, focusing on mutations, we observed that three genes were less frequently mutated in cluster A: *MUC16*, *MAST4*, and *CACNA1E*. Second, focusing on copy number alterations, we observed that amplifications of fibroblast growth factor 1 (*FGFR1*) occurred more frequently in cluster A (Pearson chi-square, *p* = 0.008) in a virtually exclusive manner with *ESR1* mutations (only 2 of 14 *ESR1* mutants also had an *FGFR1* amplification). In addition, amplifications of *TSPYL5* were enriched in cluster A (Fisher’s exact, *p* = 0.006), which occurred in only one of the *ESR1* mutated samples. Third, focusing on differential gene expression between both clusters, 17 genes were found significantly differentially expressed and with at least a two-fold change in median expression level (Mann–Whitney, FDR corrected) (Table 2).

### 3.4. Best Response on Endocrine Therapies versus Expression Clusters

Lastly, we investigated whether the obtained clusters were associated with outcome on subsequent endocrine therapy. In total, 21 patients started with endocrine therapy after their biopsy; 17 patients started with an AI, and 4 patients with fulvestrant. Exploratory analyses on best response versus the above-described clusters showed that the responses (PR, SD, and PD) were equally distributed over the two clusters (Fisher’s exact *p* = 1.00).

## 4. Discussion

To investigate the effects of DNA alterations associated with resistance against endocrine treatment on the expression of ER target genes, we here present the integration of RNA sequencing with WGS data obtained from metastatic lesions of a relatively large cohort of patients with ER-positive/HER2-negative breast cancer. Of note, we verified and addressed a bias that was discernible in the transcriptomic data, originating from the difference in the sampled biopsy sites.

We demonstrate that *ESR1* mutations, and *FGFR1* and *TSPYL* amplifications are associated with an increased expression of *ESR1* and its target genes, while mutations in genes involved in the MAPK pathway or in genes encoding ER transcriptional regulators, previously associated with endocrine resistance [8], did not correlate with increased expression of ER target genes.

Our findings on the association between *ESR1* mutations and high *ESR1* expression and its target genes adds to the compelling evidence that mutations in *ESR1* lead to constitutive activity of ER [14,37,38]. Interestingly, *FGFR1* amplifications were also enriched in samples with high ER pathway activity and occurred almost mutually exclusively with samples having an *ESR1* mutation. Our observation is in line with previous work showing that *FGFR1* amplification leads to ligand-independent ER target gene transcription [39] and mediates endocrine therapy resistance [39,40,41]. In more detail, upregulation of the growth factor *FGFR1* leads to subsequent activation of MAPK and PI3K pathways [42]. Activation of MAPK can subsequently result in estrogen-independent phosphorylation and activation of ER-α, leading to resistance to endocrine therapies such as aromatase inhibitors [43]. Hence, there is a rationale for treatment strategies combining endocrine therapy in combination with FGFR1 inhibitors in patients with an *FGFR1* amplification. Currently, a limited number of phase I and II studies on FGFR inhibitors have been conducted [44,45,46] and have shown antitumor activity in patients with *FGFR1* amplified tumors [47]. Results of phase 3 trials including only patients with *FGFR1* amplified tumors are eagerly awaited.

High expression of *TSPYL* has previously been linked to poor outcome in breast cancer patients by suppressing p53 [48]. Moreover, in a genome-wide association study, SNPs on chromosome 8 that mapped directly 3′ to the *TSPYL5* gene were associated with increased plasma estradiol concentrations in postmenopausal women eligible for adjuvant treatment with an AI following resection of an early-stage ER-positive breast cancer [49]. Further experiments showed that one of the SNPs (rs2583506) created a functional estrogen response element. However, besides suppressing p53 functionally, the exact role of *TPSYL5* amplification in *ESR1* pathway activation remains as of yet unclear.

Taking cases with *ESR1* mutations and *FGFR1/TSPYL* amplifications together, 28 out of 47 (59.6%) samples had DNA alterations associated with increased ER target gene expression, still leaving 19 samples without a recurrent genomic event that might explain the higher expression of ER target genes. The analysis of differentially expressed genes between clusters A and B did not provide additional clues why these samples had a higher ER target gene expression since most of the differentially expressed genes between clusters A and B (Table 2) are direct target genes of ER, such as *STC2* and *EGLN2*.

The observation that there is a subgroup of patients having an increased ER expression profile underlines the remaining need of blocking the ER pathway in these patients since it is very likely that these tumors still heavily rely on the ER pathway. Currently, there are interventional trials underway investigating the effectiveness of next-generation selective ER modulators and selective ER downregulators targeting both wild-type and mutant ER [50]. The question remains how to optimally select patients who will have the largest benefit from these new and existing endocrine treatment strategies. The integrative analysis of RNA sequencing and WGS in relation with response to these treatments might reveal a benefit for most of the patients in cluster A.

In contrast with *ESR1* mutations and *FGFR1* amplifications, *NF1* mutations and alterations in other MAPK pathway genes were not significantly enriched in the “active ER cluster”. This finding is in contrast with recent work [20] showing that *NF1* is a transcriptional co-repressor of ER-α that, once inactivated, leads to increased ER expression in cell line models. In our clinical samples we did not observe enrichment of *NF1* mutations in the cluster with increased ER expression. The number of nonsense mutations and structural variations in *NF1* in our cohort was low (n = 7). So, assessment of the relation between *NF1* mutation status and *ESR1* regulated gene expression should be assessed in a larger clinical cohort with a higher number *NF1* mutations.

Importantly, we observed that tumor biopsies of patients who were receiving endocrine therapies such as tamoxifen and AIs as last treatment before their biopsy were significantly more frequently observed in “ER-low” cluster B. We speculate that, since the average ER target gene expression in the “ER-low” cluster is at the same level as ER-negative tumors, these tumors classified as ER-positive might have turned into phenotypically ER-negative tumors. This intrinsic subtype switching has also recently been described in matched primary-metastatic samples of the AURORA program, occurring in 36% of cases [51]. One could argue that cases with low ER gene expression are not being dependent on the ER pathway anymore but that these cells could revert to ER dependent growth once endocrine treatment pressure is lifted. However, given the fact that the last given treatment is associated with lower expression of *ESR1* and its target genes, one should be aware that the ER pathway in these samples were probably still suppressed by the recently received endocrine therapy and that samples could have been wrongly classified as “ER-low”. Of 54 patients in cluster B, 16 patients received an AI and 7 tamoxifen as last treatment prior to their biopsy. Of the patients who received an AI, 8 of 16 patients were still on treatment at the time of biopsy, and, with regard to tamoxifen, one patient was still on treatment at the time of biopsy. Future studies focusing on transcriptomic analysis in breast cancer should take the timing of the tissue biopsy in relation to the half-life of the different endocrine therapies into account in order to prevent this possible bias.

Moreover, samples in the “ER-low” cluster (cluster B) do show some degree of heterogeneity in the expression of the ER target genes. A detailed look at the sub-clusters within the “ER-low” cluster shows that some samples with high *PGR* expression have an overall low expression of ER target genes. Although progesterone receptor (PR) activity has long been considered an indicator of functional estrogen response pathway [52], a small subset of primary breast cancer is classified as ER-negative/PR-positive with large studies showing an incidence of 3.4–3.8% [53,54]. A second sub-cluster within the “ER-low” cluster shows high *MYC* expression. Although known as an *ESR1* target gene, *MYC* is also often found activated via amplification. Taken together, samples in these two sub-clusters may be indeed independent of the ER pathway and switched to *PGR* or *MYC* enabled growth.

The power of our study resides in the large number of patients who underwent WGS and RNA sequencing on their tumor biopsy. However, our dataset also has limitations; we selected a cohort of metastatic breast cancer patients with an ER-positive/HER2-negative primary tumor. As standard pathology assessment such as ER and HER2 expression status has not been performed on these metastatic samples, and receptor conversion does occur in up to 35% of patients with an ER-positive primary tumor [55], we do not have factual evidence that all samples were still ER-positive/HER2-negative upon biopsy, which may have had an influence on the clusters found. Moreover, our cohort is heterogeneous regarding the number and type of prior lines of treatment received, which could have influenced the presence of specific genetic alterations at the DNA level as well as affected gene expression. Also, in the CPCT-02 study, tumor biopsies were taken prior to a next line of therapy for metastatic disease. Although most patients will have had their tumor biopsy upon disease progression of the prior line of therapy or at diagnosis of metastatic disease, there might be a small subgroup of patients who underwent a biopsy after experiencing toxicity. Importantly, our cohort could comprise a mix of patients who are either still sensitive or already resistant to endocrine therapies. Expanding the number of patients with similar clinical prognostic factors, such as number and type of previous therapies and number of metastatic sites, allows for more reliable association analysis between genomic and transcriptomic alterations and outcome.

## 5. Conclusions

In conclusion, we here show the potential of the integrative analysis of RNA sequencing and WGS and demonstrate that within the subgroup of ER-positive/HER2-negative breast cancers there are substantial differences in *ESR1* expression and its target genes. We further show associations between ER transcriptomic profiles and possible underlying DNA alterations. Given the fact that there is a subgroup of tumors with an increased expression of *ESR1* and its target genes, these tumors might still be responsive to drugs that target the ER pathway. To identify the largest possible group of patients who could benefit from existing endocrine treatments or new drugs targeting both wild-type and mutant ER, one should employ upfront DNA and RNA sequencing on metastatic tumor tissue to optimally relate treatment response to genomic and transcriptomic profiles.

## Figures and Tables

**Figure 1 cancers-15-04416-f001:**
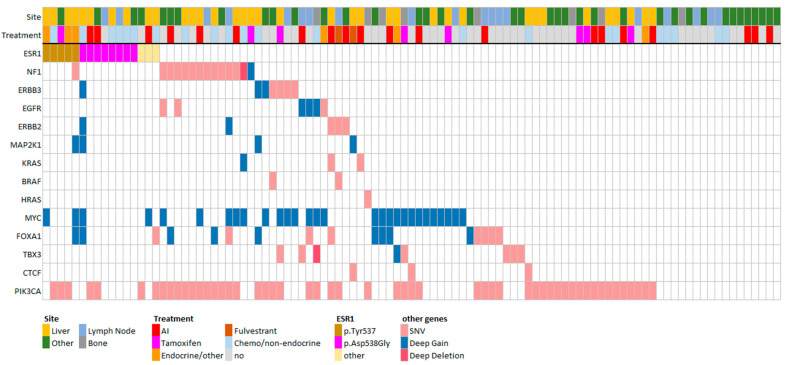
Overview of genomic alterations previously associated with endocrine resistance in our cohort. The treatment depicted in figure is the latest administered therapy prior to biopsy of the metastatic lesion. In our cohort, the following genes associated with endocrine resistance were affected: *ESR1* (n = 16), *NF1* (n = 14), *ERBB3* (n = 7), *EGFR* (n = 6)*, ERBB2* (n = 5), *MAP2K1* (n = 4), *KRAS* (n = 3), *BRAF* (n = 2), *HRAS* (n = 1), *MYC* (n = 29), *FOXA1* (n = 17)*; TBX3* (n = 8); *CTCF* (n = 3); *PIK3CA* (n = 53). Patients received the following treatments prior to biopsy: AI (n = 20), tamoxifen (n = 8), endocrine/other (n = 6), fulvestrant (n = 2), chemo (n = 22), no treatment at all (n = 43). Biopsies were obtained from the following biopsy sites: liver (n = 36), lymph node (n = 19), bone (n = 8), other (n = 38). Mutations in *ESR1* p.Tyr537 indicate either the mutation p.Tyr537Asn or p.Tyr537Ser.

**Figure 2 cancers-15-04416-f002:**
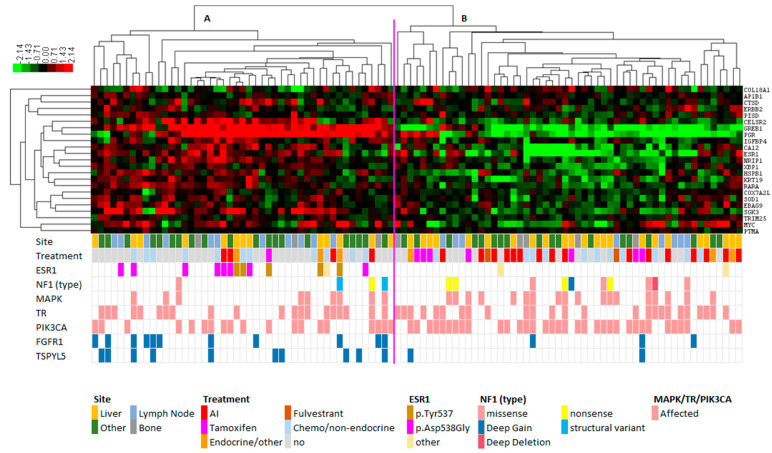
Hierarchical clustering based on ER target genes (see methods for the list of genes on which this clustering was based). Expression values were median centered per gene, with higher and lower expression than median colored as red and green, respectively. Based on the expression of ER target genes, two clusters were identified: cluster A (n = 47), characterized by a higher ESR1 expression, more *ESR1* mutations, *FGFR1* and *TSPYL5* amplifications, and cluster B (n = 54), characterized by a lower *ESR1* expression and more *PIK3CA* mutations. Mutations in *ESR1* p.Tyr537 indicate either the mutation p.Tyr537Asn or p.Tyr537Ser.

**Figure 3 cancers-15-04416-f003:**
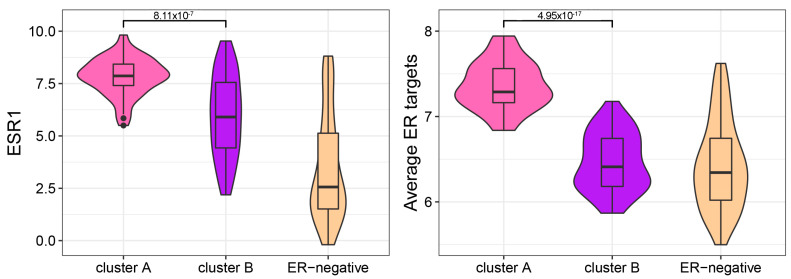
Log expression values of *ESR1* (**left panel**) and the average expression of the ER target genes (**right panel**) in samples in clusters A and B and a control group of ER-negative samples.

**Table 1 cancers-15-04416-t001:** Patient characteristics.

	Patients (n = 101)		Specification of Prior Treatments			
Age	N	%		N		%
Median (interquartile range)	59 (52–64)		Aromatase inhibitor	53		52.5
**Gender**			Tamoxifen	57		56.4
Female	101	100	Fulvestrant	16		15.8
Male	0	0	Everolimus	15		14.9
**Prior systemic therapy**			CDK4/6	6		5.9
Yes			5-FU	35		34.7
Endocrine therapy only	14	13.7	Taxanes	44		43.6
Chemotherapy only	11	10.9	Platinum/Parp	8		7.9
Endocrine and chemotherapy	51	50.5	Anthracyclines	52		51.5
Nr of lines (median, IQR)	3 (2-5)		Cyclophosphamide	50		49.5
Nr of drugs (median, IQR)	5 (4-8)		Eribulin	3		3.0
No prior treatment	25	27.8	Vinorelbine	3		3.0
**Last treatment before biopsy**			Anti-HER2	3		3.0
Tamoxifen	8	7.9				
Aromatase inhibitor	20	19.8				
Fulvestrant	2	2.0				
Combination endocrine therapy	6	5.9				
Chemotherapy or non-containing endocrine therapy	22	21.8				
No treatment within one year before biopsy	43	42.6				
**Prior radiotherapy**						
Yes	62	61.4				
No	39	38.6				
**Biopsy site**						
Liver	36	35.6				
Bone	8	7.9				
Lymph node	20	19.8				
Breast	14	13.9				
Other	7	6.9				
Unknown	3	3.0				

**Table 2 cancers-15-04416-t002:** Genes differentially expressed (beyond the ER target genes from Figure 2) between cluster A (*ESR1* wild-type) and cluster B.

Gene	*p*-Value FDR Hochberg	Fold Change *
TPBG	1.88 × 10^−8^	2.6
IGF1R	4.30 × 10^−5^	3.2
CYP2T1P	5.02 × 10^−5^	4.4
SIAH2	0.00011	2.3
FMN1	0.00039	2.1
THSD4	0.00043	3.1
AC0647992	0.00141	3.1
CUEDC1	0.00213	2.0
SUSD3	0.00402	5.3
PARD6B	0.00715	3.5
ZNF516	0.00746	2.2
PREX1	0.01012	2.6
IL6ST	0.01655	2.3
STC2	0.01764	4.6
MYBL1	0.02001	2.5
EGLN2	0.02175	2.0
COX6C	0.03034	3.5

* Median fold change cluster A over cluster B.

## Data Availability

WGS data, RNA sequencing data, and corresponding clinical data have been requested from Hartwig Medical Foundation and provided under data request number DR-068. The clinical data provided by CPCT have been locked at 30 March 2020. Both WGS and clinical data are freely available for academic use from the Hartwig Medical Foundation through standardized procedures, and request forms can be found at https://www.hartwigmedicalfoundation.nl.

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
