# Peer review of "Genomic Alterations Associated with Estrogen Receptor Pathway Activity in Metastatic Breast Cancer Have a Differential Impact on Downstream ER Signaling"

_cancers, 2023, doi:10.3390/cancers15174416_

Round 1
Reviewer 1 Report
This manuscript (MS#cancers-2559693) entitled “Genomic alterations associated with estrogen receptor pathway activity in metastatic breast cancer have a differential impact on downstream ER signaling” by Angus et al. describes the results of an analysis of mutations and gene expression patterns of relevant genes in biopsy samples (n=101) from patients with metastatic breast cancer. The authors clarified that it can be classified into two clusters and analyzed in detail the relationship with the patient background. This study provides essential information for selecting a more appropriate drug therapy for patients with metastatic breast cancer.
Please correct the following points to improve the manuscript.
Comment-1: In mutational signature analysis, the authors found that SBS2 (APOBEC mutagenesis) was enriched in cluster B. However, SBS13 is also associated with APOBEC mutagenesis. The authors should discuss why SBS13 showed a distinct pattern with SBS2.
Comment-2: “6 minutes” -> “6 min” (Page 6, lane 4 from the bottom)
Comment-3: “300bp” -> “300 bp” (Page 6, lane 3 from the bottom)
Comment-4: Regarding ESR1 mutation, “p.Tyr537Asn” would be “p.Tyr537Asn or p.Tyr537Ser”. (Figures 1, 2; Supplementary Figure 3)
Comment-5: Add Method regarding how the authors calculated the “Module score” in Legend for Supplementary Figure 2.
Comment-6: “...constitutive activity has...” -> “...constitutive activity has...” (Page 7, lane 15)
Comment-7: “table 2” -> “Table 2” (Page 8, lane 22; Page 9, lane 13 from the bottom)
Comment-8: “Figure 3” -> “Figure 3A” (Page 7, lane 3)
Comment-9: “...MAPK can...” -> “...MAPK can...” (Page 9, lane 20)
Author Response
We would like to thank the reviewer for the useful and constructive comments. We have adapted the manuscript accordingly. In the attachment is a point-by-point response to the comments of the reviewer.
Reviewer 2 Report
The authors of this study have analyzed the interplay between somatic mutations, gene expression and administered therapy in patients with ER-positive/HER2-negative metastatic breast cancer. They found two distinct clusters, one of which is associated with increased expression of ESR1 and its target genes and alterations in ESR1, FDFR1 and TSPYL, while the other group is associated with a lower ER expression comparable to ER-negative samples.
Overall, the manuscript is well written, and I only have minor comments that should be addressed.
1. In Figure 2, it could be helpful to add a box around the two clusters to help distinguish them as it is not visually clear. Also, the legend of the heatmap (what red/green represents) is missing.
2. Line 219 “As the samples in cluster B were derived from patients with ER-positive primary breast cancer”, please rephrase as all samples were derived from patients with ER+ breast cancer.
3. Figure 3. Please add the pvalues between the groups. Maybe also add them in the text.
4. Line 253: for consistency, please add the statistical test used. Only those two signatures were statistically different between the two clusters?
5. Line 386: The fact that no IHC was performed on the actual samples analyzed could potentially be an issue and should be discussed more. Indeed, some samples might have switched subtypes, as shown in other studies, and that could affect the results.
6. Lines 374-383: are these results about “ER-low” presented in the main text (please ignore this comment if they are and I missed it)?
Author Response

(The authors gave the same response as above.)
